# Meta-Analysis of the Mechanisms Underlying COVID-19 Modulation of Parkinson’s Disease

**DOI:** 10.3390/ijms241713554

**Published:** 2023-08-31

**Authors:** Jonathan Zhang, Muhammed Bishir, Sharman Barbhuiya, Sulie L. Chang

**Affiliations:** 1Institute of NeuroImmune Pharmacology, South Orange, NJ 07079, USA; zhangjon@shu.edu (J.Z.); bishirmu@shu.edu (M.B.); barbhush@shu.edu (S.B.); 2Department of Biological Sciences, Seton Hall University, South Orange, NJ 07079, USA

**Keywords:** Qiagen Knowledge Base (QKB), Ingenuity Pathway Analysis (IPA), network meta-analysis (NMA), Parkinson’s disease (PD), severe acute respiratory syndrome-coronavirus-2 (SARS-CoV-2), coronavirus disease-2019 (COVID-19), neuroinflammation, synuclein alpha (*SNCA*), angiotensin-converting enzyme-2 (ACE2)

## Abstract

Coronavirus disease-19 (COVID-19) is caused by the infection of severe acute respiratory syndrome-coronavirus-2 (SARS-CoV-2). The virus enters host cells through receptor-mediated endocytosis of angiotensin-converting enzyme-2 (ACE2), leading to systemic inflammation, also known as a “cytokine storm”, and neuroinflammation. COVID-19’s upstream regulator, interferon-gamma (IFNG), is downregulated upon the infection of SARS-CoV-2, which leads to the downregulation of ACE2. The neuroinflammation signaling pathway (NISP) can lead to neurodegenerative diseases, such as Parkinson’s disease (PD), which is characterized by the formation of Lewy bodies made primarily of the α-synuclein protein encoded by the *synuclein alpha* (*SNCA*) gene. We hypothesize that COVID-19 may modulate PD progression through neuroinflammation induced by cytokine storms. This study aimed to elucidate the possible mechanisms and signaling pathways involved in COVID-19-triggered pathology associated with neurodegenerative diseases like PD. This study presents the analysis of the pathways involved in the downregulation of ACE2 following SARS-CoV-2 infection and its effect on PD progression. Through QIAGEN’s Ingenuity Pathway Analysis (IPA), the study identified the NISP as a top-five canonical pathway/signaling pathway and *SNCA* as a top-five upstream regulator. Core Analysis was also conducted on the associated molecules between COVID-19 and *SNCA* to construct a network connectivity map. The Molecule Activity Predictor tool was used to simulate the infection of SARS-CoV-2 by downregulating IFNG, which leads to the predicted activation of *SNCA*, and subsequently PD, through a dataset of intermediary molecules. Downstream effect analysis was further used to quantify the downregulation of ACE2 on *SNCA* activation.

## 1. Introduction

The infection of severe acute respiratory syndrome-coronavirus-2 (SARS-CoV-2) causes coronavirus disease-2019 (COVID-19), which leads to inflammation primarily in the respiratory system. The recent COVID-19 pandemic has posed a significant global health risk because of its high rate of transmission and risk to immunocompromised individuals with underlying medical conditions. According to the World Health Organization’s COVID-19 Dashboard, there have been 462,758,117 confirmed cases and 6,056,725 deaths as of 17 March 2022. Furthermore, the WHO reports that, of these deaths, 1,719,002 were individuals 65+ years of age, which constitutes roughly 28.4% of all reported COVID-19 deaths.

SARS-CoV-2 enters the host cell by binding to the transmembrane receptor ACE2, where the virus–protein complex is internalized and ACE2 is subsequently downregulated [1]. As shown in Figure 1, inside the cell, SARS-CoV-2 releases its genomic RNA (gRNA), which hijacks the host’s replication and transcription complexes to produce more copies of SARS-CoV-2.

The binding of SARS-CoV-2 to ACE2 leads to the downregulation of the latter, leading to systemic inflammations also known as a “cytokine storm” [1]. The systemic inflammation then induces neuroinflammation, seen through mice models, via a nuclear-factor-kappa-B (NFκB)-induced elevation of cytokines and the subsequent inhibition of the IFNG-induced antiviral response within the brain [2]. As a result, the neurotoxicity damages the central nervous system, leading to blood–brain barrier (BBB) disruption, astrogliosis, oxidative stress, and neuronal damage and/or apoptosis [1]. Lee et al., 2022, analyzed post-mortem brain samples of COVID-19 patients. They found multifocal vascular damage accompanied by widespread endothelial cell activation. Predominant expressions of macrophages and CD8+ T cells, CD4+ T cells, and CD20+ B cells were also found in COVID-19 patients’ brains. They also reported the presence of microglial nodules in the hindbrain, suggesting neuronal loss and neurophagia [3]. These results suggest that COVID-19 infection leads to a loss in vascular integrity and a leaky BBB, which in turn results in the entry of systemic pathogens and inflammatory cytokines into the brain and triggers neuroinflammation in the various brain regions.

PD is a neurodegenerative disorder that leads to bradykinesia, tremors, and muscular rigidity [4]. It is characterized by the loss of dopaminergic (DA) neurons in the substantia nigra and the presence of Lewy bodies in neurons [5]. The major morphological change in PD brains is the loss of darkly pigmented areas in the substantia nigra pars compacta and the locus coeruleus. This is due to the loss of the DA-neuromelanin-containing neurons in the substantia nigra pars compacta and the noradrenergic DA neurons in the locus coeruleus [6]. About 30% of the DA neurons are lost during the onset of motor symptoms followed by 60% or more DA loss, which correlates with severe motor deficits and extended disease progression [7]. Another major characteristic of PD is mitochondrial dysfunction with mitochondrial complex-I deficiency, which is crucial in DA cell death due to the depletion of energy [8]. Lewy bodies are a hallmark of several neurodegenerative diseases including PD [9]. These are aggregates formed from α-synuclein protein fibers, produced by the gene *SNCA*. *SNCA* mutations may lead to an improper amino acid sequence, resulting in the misfolding of the protein’s three-dimensional structure [9]. PD pathology also consists of dysfunctional protein clearance systems, where the ubiquitin-proteasome system and the autophagy-lysosome pathway are damaged, which disrupts the clearance of monomeric α-synuclein protein, resulting in aggregation and misfolding of α-synuclein protein [10]. Neuroinflammation is a key contributor to PD pathogenesis, where microglial-induced inflammation is suggested to be from α-synuclein and causes the degeneration of the DA cells [11]. PD is a complex neurodegenerative disease with various complex pathogenic mechanisms, which are not fully understood. COVID-19 is a novel disease and paired with PD makes it especially burdensome to patients. The present study aimed to elucidate the underlying mechanisms of COVID-19’s negative modulation of PD pathology by focusing on *SNCA* expression and neuroinflammation.

A previous study by Semerdzhiev, Slav A et al. showed that the nucleocapsid protein (N-protein) in COVID-19 affects α-synuclein [12]. The SARS-CoV-2 N-protein possesses an affinity for α-synuclein, which leads to the aggregation of amyloid fibers [12]. The aggregation of amyloid fibers, also known as Lewy bodies, form within neurons [5]. The N-protein is a major structural protein that facilitates viral RNA production, suppresses intracellular immune responses, and packages viral RNA into developing virions. The N-protein consists of an N-terminal RNA binding domain (RBD) and a C-terminal dimerization domain (CTD) [12]. Electrostatic interactions play a part in N-protein and α-synuclein binding: the N-protein has a calculated net positive charge of +24e, while it has a calculated net negative charge of −9e [12]. Through multiple Thioflavin T (THT) assays and in vitro microinjection of SH-SY5Y cells with the N-protein, the literature has concluded that neuro-COVID-19 leads to the aggregation of amyloid fibers and the subsequent formation of Lewy bodies [12]. Coronavirus antibodies have been found in the CSF fluids of patients with PD [13]. Furthermore, ACE2 is highly expressed in DA neurons, and the degenerative pathology of PD decreases the number of DA neurons. This suggests that SARS-CoV-2 neuroinvasion may disrupt the dopamine pathway and cause worsening symptoms in patients with PD [14]. We hypothesize that SARS-CoV-2 viral exposure triggers *SNCA* activation and neuroinflammation in the brains of COVID-19 patients, which in turn results in neurodegeneration and the progression of PD pathology.

The mechanisms underlying COVID-19 augmentation of PD are still unknown. Patients develop PD, which is a common neurodegenerative disease, at the late stages of their lives, usually after 60. However, about 5–10% of PD patients show early onset, below 50. COVID-19 initiates peripheral inflammatory responses. Systemic inflammation can lead to neuroinflammation and neurodegeneration. It is crucial to understand whether COVID-19 patients will develop PD symptoms in the future. Some reports also suggest that SARS-CoV-2 infection and its long-term complications result in premature aging [15,16]. A review article by Leta et al. examined several studies showing that people with PD had worsening symptoms during the acute phase of SARS-CoV-2 infection [17]. In a case–control study of 16,971 patients, PD patients with SARS-CoV-2 infection had worsening motor (63%) and non-motor (75%) symptoms compared to PD patients without SARS-CoV-2 (43% worsening motor symptoms and 52% non-motor symptoms) [18]. This suggests that COVID-19 worsens PD pathology. Leta et al. hypothesized that SARS-CoV-2 infection causes stress and inflammation, altered dopaminergic neurotransmission, COVID-19 restriction and social isolation, and SARS-CoV-2-induced specific damage, which makes PD patients more vulnerable to the SARS-CoV-2 infection [17]. There have been 20 case studies as of 2022 that have shown new onset of parkinsonism symptoms such as bradykinesia and/or rigidity during or shortly after the diagnosis of COVID-19 [17]. Furthermore, studies have shown that the suggested worsening of the pathology of PD patients may be due to viral post-encephalitic parkinsonism, while others suggest that SARS-CoV-2 neuroinvasion may be responsible [17,19,20].

The existing studies suggest a possible association between COVID-19 and PD. Hence, understanding the commonly shared molecules including genes, proteins, and their associated signaling pathways that can potentially contribute to the development of PD in SARS-CoV-2 infections is critical. Over the last few decades, bioinformatics network meta-analysis has received greater attention among scientists to identify the key molecules and signaling pathways involved in various pathological conditions. In this study, we employed IPA, which is a bioinformatics tool that employs machine learning and network algorithms to identify the relationships among the molecules, signaling pathways, upstream regulators, diseases, and biological functions, as well as their predicted activation state. IPA uses the Qiagen Knowledge Base (QKB) to power its data analysis and interpretation. The QKB is a repository of over seven million findings from scientific literature. The QKB also has manually designed signaling pathways that have been created by scientists based on the findings in the QKB. IPA’s algorithm employs a right-tailed Fisher exact test to examine and predict the involvement of a signaling pathway in the uploaded set of molecules. Our meta-analyses identified the commonly shared molecules associated with COVID-19 and PD, as well as signaling pathways and upstream regulators associated with the shared molecules. We used the IPA tools to reveal the downstream effects of simulated SARS-CoV-2 infection via the downregulation of IFNG. Our meta-analyses suggested that COVID-19 elevates *SNCA* expression with the possible aggregation of α-synuclein and the formation of Lewy bodies through the NISP, which would subsequently enhance neurodegeneration, leading to the augmentation of PD onset and progression.

## 2. Results

### 2.1. Upstream Regulators and Signaling Pathways Involved in COVID-19 Influencing PD 

Figure 2a lists the various IPA tools used to identify the signaling pathways and upstream regulators. We used IPA’s “Grow” tool to expand and collect the molecules and chemicals associated with COVID-19 and PD, respectively. There were 830 molecules associated with COVID-19 and 559 molecules associated with PD. As shown in Figure 2b, using IPA’s “Compare” tool, we identified 84 overlapping molecules between the molecules associated with COVID-19 and PD. Three of these eighty-one overlapping molecules were chemical and biological drugs that are not naturally found in the human body, so they were removed, leaving eighty-one overlapping molecules for subsequent analyses. We used IPA’s “Core Analysis” tool to identify the upstream regulators and signaling pathways involved in these 81 overlapping molecules. Our Core Analysis revealed the NISP to be one of the top signaling pathways involved, as shown by Figure 3a. We also identified *SNCA* as the top upstream regulator associated with COVID-19 and PD, as shown by Figure 3b.

### 2.2. Upstream Regulators and Signaling Pathways Involved in COVID-19 Influencing SNCA Expression

Core Analysis of the overlapping molecules associated with COVID-19 and PD revealed *SNCA* as one of the top upstream regulators, as shown in Figure 3b. As shown in Figure 4a, we used IPA’s “Grow” tool to expand the molecules and chemicals that were associated with COVID-19 and *SNCA*. There were 830 molecules and chemicals grown from COVID-19 and 1217 molecules grown from *SNCA*. Using IPA’s “Compare” tool, we identified 101 overlapping molecules between COVID-19 and *SNCA*. Nine of the one-hundred and one overlapping molecules were chemical and biological drugs that are not naturally found in the human body, so they were removed, leaving ninety-two overlapping molecules, as shown in Figure 4b. IPA’s “Core Analysis” tool was used to identify the signaling pathways involved in the 92 overlapping molecules. The NISP was identified to be the top among the signaling pathways associated with COVID-19 and *SNCA*. Other top signaling pathways included glucocorticoid receptor signaling and the role of hypercytokinemia/hyperchemokinemia in the pathogenesis of influenza, as shown in Figure 5.

### 2.3. Pathway Connectivity Map to Examine COVID-19 Modulation of SNCA

Using the 92 overlapping molecules associated with COVID-19 and *SNCA*, we constructed a connectivity map between COVID-19, overlapping molecules, and *SNCA*. The literature and IPA’s Core Analysis of molecules associated with COVID-19 identified IFNG as the top upstream regulator upon downregulation in SARS-CoV-2 infection. IFNG was manually added to the connectivity map. We added ACE2, an important receptor responsible for SARS-CoV-2 internalization into the cell and subsequent signal transduction cascades, to the connectivity map. Relationships were drawn from COVID-19 to IFNG and IFNG to ACE2 and, then, to the overlapping molecules using IPA’s “Connect” and “Pathway Explorer” tools. Similarly, the relationships from overlapping molecules to *SNCA* and PD were also drawn. We then employed IPA’s molecule activity predictor (MAP) tool to simulate the inhibition of IFNG to mimic COVID-19, which downregulated ACE2 expression. The molecules that did not show any simulated effects were removed from further analysis. The concurrent downregulation of IFNG and ACE2 activated the expression of 20 molecules and inhibited the expression of 14 molecules, eventually leading to an increase in *SNCA* expression and PD pathogenesis, as shown in Figure 6a. Core Analysis was conducted on the 34 molecules involved in the overall activation of *SNCA* and PD from the downregulation of IFNG. Signaling pathways were identified using Core Analysis. As shown in Figure 3a and Figure 5, the NISP was identified to be a significantly enriched signaling pathway among the top signaling pathways identified in the Core Analysis of COVID-19’s influence on PD and COVID-19’s influence on *SNCA*, respectively. “Cytokine storms” were also identified to be the topmost upregulated signaling pathway.

### 2.4. Quantitative Analysis of Upstream Regulators in COVID-19 Modulation of SNCA

To observe the overall impact of COVID-19 on *SNCA*, the 34 molecules involved in the overall activation of *SNCA* and PD were analyzed using downstream effect analysis. As shown in Figure 7, interleukin-1 β (IL1B), c-c motif chemokine ligand 5 (CCL5), and angiotensin (AGT) were used as intermediary molecules to identify the strength of the influence of ACE2 on *SNCA*. The overall z-score was −0.43229.

To quantify the influence of ACE2 on *SNCA* and expand the network of associated molecules, each of the intermediary molecules was grown using IPA’s “Grow” tool. These molecules were trimmed using IPA’s “Trim” tool to keep only the associated molecules that had a downstream effect of ACE2 and an upstream effect on *SNCA*. The pathway maps were constructed from IL1B’s, CCL5’s, and AGT’s associated molecules using IPA’s “Pathway Explorer” tool.

To begin with, IL1B was grown using IPA’s “Grow” tool to identify all the associated molecules and then connected to ACE2 via IPA’s “Connect” tool. AGT, extracellular signal-regulated kinase 1/2 (ERK1/2), matrix metallopeptidase 2 (MMP2), CCL5, and matrix metallopeptidase 9 (MMP9) showed downstream effects of ACE2 and upstream effects on *SNCA*. A network was constructed from IL1B and its molecules associated with ACE2 and *SNCA*. Simulated inhibition of ACE2 showed a potential increase in the expression of *SNCA* with a z-score of −0.683, as shown in Appendix A. The same workflow was applied to CCL5, and a network was constructed between ACE2, the associated molecules of CCL5, and *SNCA*, as shown in Appendix A. The simulated inhibition of ACE2 activated the associated molecules of CCL5 (IL1B, ERK1/2, MMP2, and MMP9), leading to an increase in *SNCA* expression. The z-score for the ACE2-downregulation-mediated increased expression of *SNCA* via CCL5’s associated molecules was found to be −0.946. Similarly, ACE2 inhibition in the network constructed between ACE2, the associated molecules between AGT, and *SNCA* increased the expression of *SNCA*. Simulated inhibition of ACE2 increased the expression of ERK1/2, MMP9, IL1B, and MMP2 and decreased the expression of CCL5, leading to an overall activation of *SNCA*, as shown in Appendix A. The z-score for the AGT-mediated increase in *SNCA* expression was found to be −0.683. The overall z-score for the involvement of IL1B, CCL5, and AGT in the ACE2-downregulation-mediated increased *SNCA* expression was found to be −2.312, corresponding to a *p*-value of 0.0706 of a two-tailed distribution at a 95% confidence interval. Downstream effect analysis provided a z-score for each of the following intermediary molecules involved in ACE2’s influence on *SNCA*, as shown in Figure 8. AGT was the intermediary molecule with the greatest influence on ACE2’s effect on *SNCA* with a z-score of 0.97 followed by CCL5 with a z-score of 0.57 and IL1B with 0.68.

### 2.5. Pathway Connectivity Map to Examine COVID-19 Modulation of the NISP

The previous results concluded that COVID-19 modulates *SNCA* expression. Next, we studied the influence of the overlapping molecules with the NISP and the involvement of the NISP in COVID-19 modulation of PD using a pathway connectivity map, as shown in Figure 9. The NISP was added to the pathway constructed in Section 2.3. Using IPA’s “Connect” and “Pathway Explorer” tools, we implemented relationships from the NISP to other nodes onto the pathway. Molecules that did not have relationships with the NISP were removed.

### 2.6. Quantitative Analysis of Upstream Regulators in COVID-19 Modulation of the NISP

To observe the overall impact of COVID-19 on the NISP, the 12 molecules involved in the overall activation of the NISP were analyzed using downstream effect analysis. As shown in Figure 10, six molecules, c-c motif chemokine ligand 2 (CCL2), CCL5, prostaglandin-endoperoxide synthase 2 (PTGS2), IL1B, tumor necrosis factor (TNF), and interleukin 6 (IL6), were used as intermediary molecules in downstream effect analysis.

Similar to downstream Effect analysis, as shown in Section 2.4, each of the intermediary molecules had a network of associated molecules that were grown and trimmed down to have a downstream effect on ACE2 and an upstream effect on the NISP. The pathway maps were constructed from each of the intermediary molecules from Figure 10. IPA’s “Grow” tool and “Connect” tool were used to connect ACE2 and the NISP to each of the grown molecules of the intermediary molecules. The following constructed pathways of the intermediary molecules CCL2, CCL5, PTGS2, IL1B, TNF, and IL6, are shown in Appendix A, respectively. The z-score was computed for each of the intermediary molecules involved in ACE2-downregulation-mediated increased expression of the NISP via downstream effect analysis as shown in Figure 11. CCL5 had the greatest influence on ACE2’s effect on the NISP with a z-score of −1.10, and PTGS2 had the least influence with a z-score of −0.34. The other intermediary molecules had the following z-scores of −0.47, −0.64, −0.68, and −0.97 for TNF, IL1B, CCL2, and IL6, respectively.

## 3. Discussion

The present network meta-analysis study suggests a strong association between COVID-19 in the progression of PD. Our results suggest that COVID-19-mediated downregulation of the ACE2 receptor increases *SNCA* expression. Molecules associated with both COVID-19 and PD were analyzed through Core Analysis, which showed the NISP as one of the top signaling pathways involved within both diseases with a *p*-value of 3.1 × 10^−14^. This suggests that the NISP is highly involved in COVID-19 and PD. The literature shows that COVID-19 causes cytokine storms, which lead to neuroinflammation and, subsequently, neurodegenerative diseases. To further examine the impact of COVID-19 on PD through the NISP, *SNCA* was chosen as a molecule of interest. Proteins synthesized from *SNCA* aggregate to form Lewy bodies, which are deposited in the neurons of various brain regions resulting in PD. The Core Analysis of the overlapping molecules associated with both COVID-19 and PD revealed *SNCA* as a top upstream regulator (*p*-value of 2.25 × 10^−16^), suggesting the involvement of *SNCA* in the pathology of COVID-19 and PD. The main aim of the present study was to determine how COVID-19 affects *SNCA* and how ACE2 expression modulates PD pathology.

COVID-19 patients exhibit a significant increase in plasma cytokine levels, including interleukins (IL1B, IL1RA, IL7, IL8, IL9, IL10), fibroblast growth factor, granulocyte colony-stimulating factor, granulocyte-macrophage colony-stimulating factor, IFNG, IP10, monocyte chemoattractant protein 1, macrophage inflammatory proteins (MIP1A and MIP1B), platelet-derived growth factor, TNF, and vascular endothelial growth factor (VEGF), compared to a healthy control group [21]. Hamster models infected with SARS-CoV-2 and patients who died from COVID-19 showed impaired BBB permeability, microglial activation, and increased brain expression of IL-1β and IL6 within the hippocampus and the inferior olivary nucleus of the medulla when compared to the control groups. These alterations result in neuroinflammation, loss of hippocampal neurogenesis, and neurocognitive symptoms [22]. Our results were consistent with the previous findings. Core Analysis conducted on the associated molecules in both COVID-19 and PD identified the topmost significant signaling pathway as the NISP with a *p*-value of 1.72 × 10^−16^. This means that the NISP plays a crucial role in both COVID-19 and PD. The third-most-significant signaling pathway is cytokine storms with a *p*-value of 2.51 × 10^−14^. Cytokine storms are the body’s immune response, causing the release of inflammatory proteins called cytokines. Infection by SARS-CoV-2 causes the release of cytokine storms. This further supports the theory that COVID-19 causes cytokine storms, which then cause neuroinflammation.

To identify the impact of the associated molecules identified in both COVID-19 and *SNCA*, we placed these molecules into a pathway map, where IFNG was downregulated to simulate infection by SARS-CoV-2. IFNG is the topmost upstream regulator in the molecules associated with COVID-19. IFNG is a cytokine that is significantly downregulated upon SARS-CoV-2 infection during the cytokine storm. The simulated downregulation of IFNG using IPA’s MAP tool showed the inhibition of ACE2. The receptor-mediated endocytosis of SARS-CoV-2 via ACE2 leads to the downregulation of ACE2. Thirty-four molecules from the set of molecules associated with COVID-19 and *SNCA* were consequently affected by the inhibition of ACE2. Of those 34 molecules affected, 14 molecules were inhibited and 20 molecules were activated. These molecules lead to an overall activation of *SNCA*, which subsequently leads to the activation of PD.

To quantify ACE2’s impact on the activation of *SNCA*, we conducted downstream effect analysis. The downstream effect analysis of the 34 intermediary molecules downstream of ACE2 and upstream of *SNCA* quantified the effects that COVID-19 had on PD. Downstream effect analysis was performed based on the literature findings from the QKB, where ACE2 impacted the intermediary molecule’s expression and where the intermediary molecules impacted the expression of *SNCA*. Three molecules (IL1B, CCL5, and AGT) were identified and used for analysis for the downstream effect analysis. Downstream effect analysis would yield a large margin of error given the sample size of three when used for the analysis, so the associated molecules of each intermediary molecule with downstream effects of ACE2 and upstream effects on *SNCA* were identified. Downstream effect analysis was conducted on the associated molecules of the intermediary molecules. The z-score was computed for each of the intermediary molecules showing IL1B with a z-score of 0.682689, CCL5 with a z-score of 0.56803, and AGT with a z-score of 0.965468. The upregulation of pro-inflammatory cytokine IL1B and the downregulation of neuroprotective agent CCL5 from the inhibition of ACE2 play a role in the neurotoxicity associated with the neuroinflammation response. ACE2 is responsible for converting angiotensin II into angiotensin (1–7). The inhibition of ACE2 causes the accumulation of angiotensin II, which is encoded by the gene AGT. Angiotensin II receptor type 1 (AGTR1) converts excess angiotensin II, which leads to inflammation, thrombosis, oxidative stress, and fibrosis [23]. Upon receptor-mediated endocytosis of angiotensin II from AGTR1, vasoconstriction leads to hypertension, contributing to neuronal injury, which causes neurodegeneration associated with PD and Alzheimer’s disease [24]. The overall *p*-value of these intermediary molecules was less than 0.05, which shows a significant impact of the downregulation of ACE2 on the activation of *SNCA*. This shows that IFNG downregulation mimicking SARS-CoV-2 infection leads to a significant impact on the activation of *SNCA*, leading to PD, as shown in Figure 7.

After identifying the NISP’s involvement in COVID-19’s activation of PD, the associated molecules between COVID-19 and *SNCA* were trimmed to only the molecules that were involved in the NISP. These molecules were further trimmed to identify those molecules downstream of ACE2 and upstream of the NISP, leaving six molecules. IPA’s “Grow” tool was used on these six molecules (CCL2, CCL5, PTGS2, IL1B, TNF, IL6) to identify the associated molecules for the downstream effect analysis. These molecules had a significant overall *p*-value of less than 0.05. Chemokine receptors, specifically CCL5, have multiple functions in the CNS, including neuroinflammation, insulin signaling, neuromodulation of synaptic activity, and neuroprotection against a variety of neurotoxins [25,26]. Avdoshina et al. showed that morphine administration enhances the release of CCL5 from astrocytes that inhibit HIV gp120 BaL-mediated neurotoxicity [25]. In addition, CCL5 has also shown neuroprotective activity against various neurotoxins including viral proteins gp120 [27], the transactivator of transcription (TAT) [28], glutamate [29], and β-amyloid [30]. The neuroprotective properties of CCL5 are mainly attributed to its potential to increase neurotrophic factors, such as brain-derived neurotrophic factor and epidermal growth factor [31]. Our results suggest that COVID-19-mediated downregulation of IFNG inhibits CCL5 expression, as shown in Figure 10. Along with the upregulation of inflammatory mediators such as IL1B, TNF, and IL6, the downregulation of CCL5 can be a major contributor to COVID-19-mediated neuroinflammation. PTGS2 codes for an enzyme called cyclooxygenase 2 (COX-2), which contributes to neuroinflammation. Increased expression of COX-2 in activated microglial cells can increase the production of pro-inflammatory cytokines such as TNF and IL1B. These pro-inflammatory cytokines contribute to the neuroinflammation response [32]. The present network’s meta-analysis study suggests that COVID-19-induced downregulation of ACE2 elevates *SNCA* expression through the NISP, leading to the formation and aggregation of Lewy bodies and eventually leading to PD progression. The downstream effect analysis of ACE2’s effect on *SNCA* expression showed significant overall activation of PD from COVID-19.

The bioinformatics tool, IPA, gathers millions of literature findings curated by thousands of scientists to create connections that are time- and material-efficient compared to other methods of gathering information. COVID-19 is a global pandemic and a large health concern for the past few years that has taken millions of lives and infected many more. To study how this global pandemic affects patients with PD, using IPA is a time-efficient tool that allows us to examine much literature and simulate predictive infections of SARS-CoV-2 to examine the PD pathways that have been studied before. By using IPA, we can study a new disease’s impact on a well-studied disease and predict the impact of the new disease on a well-studied diseased, which we have done with COVID-19 and PD. Due to the novel nature of COVID-19, its effect on PD is not well studied. Thus, the strength of IPA allows us to view the associated molecules from the literature on COVID-19 to provide us with more insight into COVID-19’s impact. Furthermore, PD is a disease that develops over time, which makes researching the long-term effects of COVID-19 on PD progression difficult to examine. Using IPA’s MAP tool to simulate the infection of SARS-CoV-2 and examining its effect on the pathways involved in the progression of PD, we can predict the effects of COVID-19 on PD without needing to perform long-term studies on populations that have been infected with COVID-19 and their development of PD.

We used IPA’s MAP tool to simulate the infection of SARS-CoV-2, which allowed us to identify and focus on molecules related to PD that have been impacted by COVID-19. Identifying these molecules and using the upregulation and downregulation tools to simulate the changes of expression in molecules can be used to identify drug targets by simulating changes where *SNCA* and molecules involved with PD progression, as well as the NISP are downregulated. The downregulation of ACE2 shows an increase in the NISP, as well as *SNCA* and, subsequently, PD. ACE2 is a potential drug target, where upregulation of ACE2 would help alleviate the neuroinflammation associated with COVID-19, as well as the activation of *SNCA* and subsequent PD progression.

### Limitations

As in other experimental methodologies, there are potential limitations that could impact the results of our meta-analysis. First, the molecules identified in the QKB that are influenced by COVID-19 are curated from the literature, which may not encompass all other studies that do not meet the criteria. The search filters and QKB database include literature before April 2022. Secondly, the information in our database does not sort for vaccination status, so both vaccinated and unvaccinated literature datasets were handled with the same weight. Third, IPA uses overrepresentation analysis, where the statistical significance of a particular pathway is calculated by evaluating the ratio of differentially expressed components within the pathway, rather than a proportion that would be expected by chance. This assigns equal weight to each pathway component irrespective of the intrinsic significance of its interactions. Fourth, our study has scarce data regarding COVID-19 patients that may develop PD in the future, so evaluating the effects of COVID-19 on PD progression presents a challenge. However, it is important to note that PD can take several decades to progress, and COVID-19 is a novel disease. Thus, case-controlled studies are needed to examine the effect of COVID-19 on PD progression and severity. Fifth, IPA generates predicted effects that are simulated based on a computational algorithm. Therefore, to understand the exact biological mechanisms and pathways of COVID-19-induced increased severity or progression of PD, direct mechanistic studies are warranted. Nevertheless, our meta-analyses identify key molecules and pathways of interest that could be possible significant contributors to COVID-19 augmentation of PD pathologies.

## 4. Materials and Methods

### 4.1. Ingenuity Pathway Analysis Software

The IPA Analysis Match CL license was purchased from QIAGEN for using all features and tools of the IPA 22.0 software. All data used for the COVID-19 and PD analysis for this study were accessed and retrieved from the QKB between 12 January 2022 and 24 April 2022 (QIAGEN Inc., Hilden, North Rhine-Westphalia, Germany https://www.qiagenbioinformatics.com/products). The QKB is a database of over 12 million literature findings from about 3600 peer-reviewed journals compiled by over 2000 QIAGEN scientists. These literature findings are experimentally demonstrated, not just from abstracts. Furthermore, information from third-party databases such as BioGRID, ClinGen, and PubChem are manually reviewed and incorporated into the QKB. Experimental data are taken from the literature articles, where the interactions are between molecule and molecule, molecule and cell and cellular process, molecule and disease, molecule and tissue, and molecule and phenotype. The molecule refers to any gene, RNA, protein, or chemical. The database covers mammalian information from humans, mice, and rat genes.

### 4.2. Identification of Associated and Overlapping Molecules for Pathway Construction

IPA’s “Grow” tool produces a network of all genes, proteins, complexes, and chemicals affected by the selected molecule. In the present study, COVID-19, PD, and *SNCA* were all expanded through IPA’s “Grow” tool. These include genes, proteins, chemicals, and complexes affected by the respective nodes identified from the QKB. Chemical drugs and toxicants that do not occur in biological systems were eliminated from further analysis using the “Trim” tool. IPA’s “Path Explorer” tool and IPA’s “Connect” tool were used to join the overlapping molecules associated with COVID-19, PD, and *SNCA* together to construct a pathway based on relationship findings found in the QKB. The “Connect” tool connects specific known molecular interactions among the other nodes in the pathway. The “Path Explorer” tool generates the shortest path or molecular connections between an upstream molecule, disease, or function of interest to a downstream molecule, disease, or function of interest based on the relationships and findings in the QKB. In the present study, we used the “Path Explorer” tool to generate molecular connections between IFNG, overlapping molecule of *SNCA* and PD, and *SNCA*.

### 4.3. Identification of Upstream Regulator and Signaling Pathways via Core Analysis

The molecules identified to overlap in each of the networks were compiled into a dataset that was then uploaded to IPA for separate Core Analysis. Core Analysis revealed the top upstream regulators and top signaling pathways associated with the molecules. The significance of the upstream regulators and signaling pathways was calculated using the Benjamini–Hohenberg-corrected Fischer’s exact test to generate the –log(*p*-value) for multiple hypothesis test correction. The significance of the canonical pathway *p*-value calculated from the right-tailed Fischer’s exact test is the *p*-value of overlap, which represents the probability of association of molecules from the dataset with the canonical pathway by random chance alone. These canonical pathways are well-characterized metabolic and signaling pathways that are curated and hand-drawn by Ph.D.-level scientists based on pathways from peer-reviewed journal articles, review articles, textbooks, and HumanCyc for metabolic pathways. Upstream regulators were identified from the dataset based on the information from the QKB in which the regulator shows the observed changes identified in the dataset. The *p*-value of overlap of the upstream regulator represents the significance of genes in the dataset that are downstream of the regulator.

### 4.4. Quantitative Analysis of COVID-19 on SNCA Expression and the NISP

IPA shows the qualitative relationships when IPA’s MAP tool is used to predict the upstream and/or downstream effect of the simulated activation or inhibition of a molecule. The “unknown” molecules that are not manually simulated by activation or inhibition by the “known” simulated molecule have a z-score calculated based on the relationship findings from the QKB from the neighboring “known” simulated molecules that are connected to it. The predicted activity pattern is calculated by the z-score of these “unknown” molecules by using the z-score error function: erf (z-score/root 2), where a value close to 1/−1 is a high/low z-score and a value that is close to 0 is a weak z-score. This is used to measure the confidence of the predicted activity with 1 and −1 being the highest possible confidence value. The values of these scores are visualized in the pathway maps as blue- and orange-colored nodes, where the blue nodes represent predicted inhibition and the orange nodes represent predicted activation. A darker shade of blue or orange represents more confidence in the predicted inhibition and activation, respectively; likewise, a lighter shade of color represents less confidence in the predicted activity.

To determine the quantitative impact of these qualitative relationships found using the MAP tool, downstream effect analysis was performed as described previously [20,33,34,35,36]. The analysis determines the significance of a molecule’s effect on either another molecule or a signaling pathway by quantifying the relationship between the nodes on IPA. The formula computes a z-score for each molecule that is downstream of the selected upstream target. The z-score ranges between −2 and +2; −2 indicates a strong inhibitory relationship, and +2 indicates a strong activation. Individual z-scores were then combined to obtain the “local z-score”, which corresponds to a z-statistic of a normal distribution.

## 5. Conclusions

The present study was designed using IPA to understand the mechanisms by which COVID-19 affects PD. A set of interacting molecules was identified between COVID-19 and PD, as well as COVID-19 and *SNCA*, where Core Analysis identified the top signaling pathways and upstream regulators. This prompted a pathway to be constructed between COVID-19 and *SNCA* with IFNG, ACE2, and intermediary molecules consisting of cytokines, receptors, and enzymes. From this, we ran a Core Analysis on the pathway and identified “cytokine storm” to be the top signaling pathway followed by the NISP. To further explore the role of the NISP in COVID-19’s modulation of PD, the previously constructed pathway was altered to focus on molecules associated with the NISP. When IFNG was downregulated to mimic SARS-CoV-2 infection, ACE2 was downregulated and a set of intermediary molecules was affected, leading to an overall activation of *SNCA* and subsequent activation of PD. This confirmed our theory that COVID-19 causes inflammation, which leads to neuroinflammation and, eventually, neurodegenerative diseases such as PD. Through downstream effect analysis, the relationship between molecules was analyzed to provide insight into the statistical significance of COVID-19’s effects within the constructed pathways leading to PD. Key molecules identified in the pathway revealed potential therapeutic targets to mitigate the progression of PD induced by COVID-19’s neuroinflammation.

## Figures and Tables

**Figure 1 ijms-24-13554-f001:**
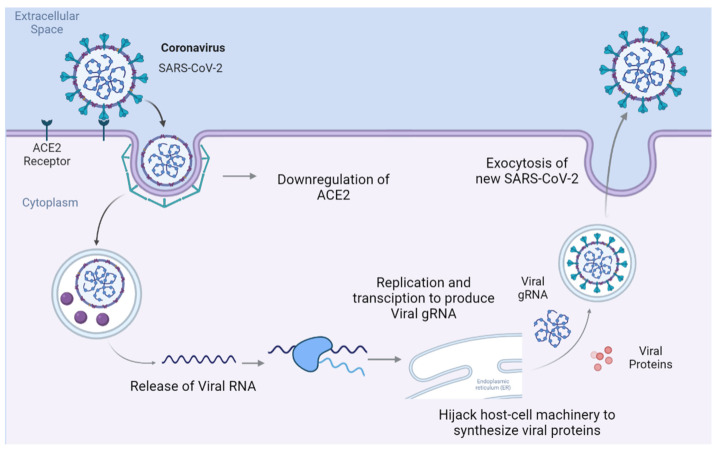
Overview of SARS-CoV-2 life cycle from entry into a host cell via receptor-mediated endocytosis to the release of replicated SARS-CoV-2. The receptor-mediated endocytosis of SARS-CoV-2 leads to the downregulation of ACE2. Once in the cytoplasm, the virus hijacks the host cell’s machinery to synthesize viral proteins and RNA to replicate itself, after which it spreads to other cells.

**Figure 2 ijms-24-13554-f002:**
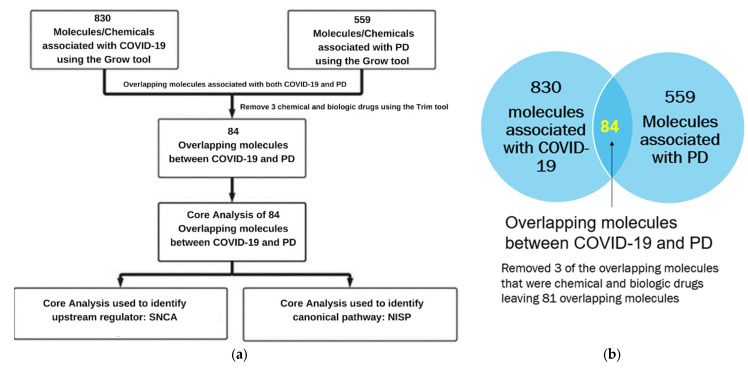
(**a**) Workflow diagram to identify the upstream regulators and signaling pathways involved in COVID-19 and PD. The molecules associated with COVID-19 and PD were obtained using the IPA “Grow” tool. The overlapping molecules obtained from IPA’s “Compare” tool were analyzed using IPA’s “Core Analysis” tool to identify the upstream regulators and signaling pathways. (**b**) Venn diagram of molecules associated with COVID-19 and PD identified from the QKB. IPA’s “Compare” tool was employed to identify the overlapping molecules associated with COVID-19 and PD. We identified 84 overlapping molecules common between the two sets of molecules, and 3 molecules that were chemical and biological drugs were removed from further analysis.

**Figure 3 ijms-24-13554-f003:**
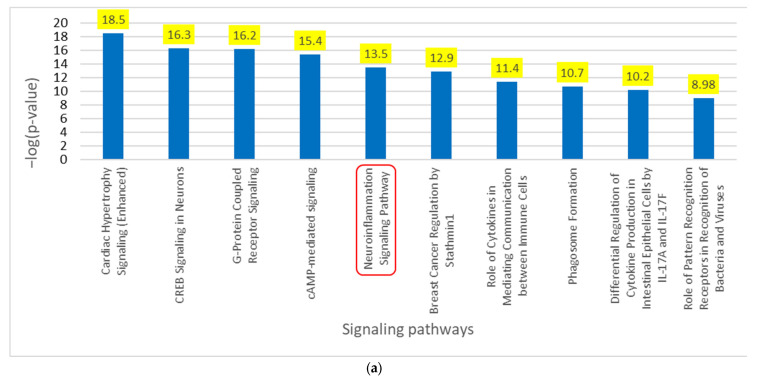
(**a**) Core Analysis of the top 10 enriched canonical pathways involved in overlapping molecules between COVID-19 and PD. Core Analysis of the overlapping molecules associated with COVID-19 and PD revealed the NISP (*p*-value of 3.1 × 10^−14^) as one of the top enriched signaling pathways as noted in the red square. (**b**) Core Analysis of the top 10 upstream regulators involved in overlapping molecules between COVID-19 and PD. Core Analysis of the overlapping molecules associated with COVID-19 and PD found *SNCA* (*p*-value of 2.25 × 10^−16^) as one of the top 5 upstream regulators as shown in red.

**Figure 4 ijms-24-13554-f004:**
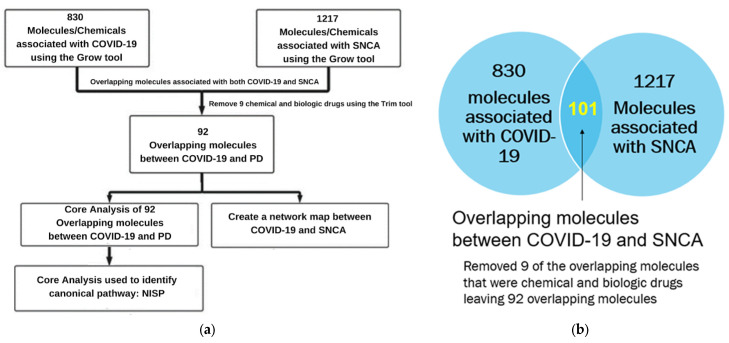
(**a**) Workflow diagram to identify the upstream regulators and signaling pathways involved in COVID-19 and *SNCA*. Using IPA’s “Grow” tool, we identified molecules associated with COVID-19 and *SNCA*. We then analyzed the overlapping molecules obtained from IPA’s “Compare” tool using IPA’s “Core Analysis” tool to identify the upstream regulators and signaling pathways. (**b**) Venn diagram of molecules associated with COVID-19 and SNCA identified from the QKB. We employed IPA’s “Compare” tool to identify the overlapping molecules associated with COVID-19 and *SNCA*. We identified 92 overlapping molecules between COVID-19 and *SNCA* and removed 3 molecules that were chemical and biological drugs from further analysis.

**Figure 5 ijms-24-13554-f005:**
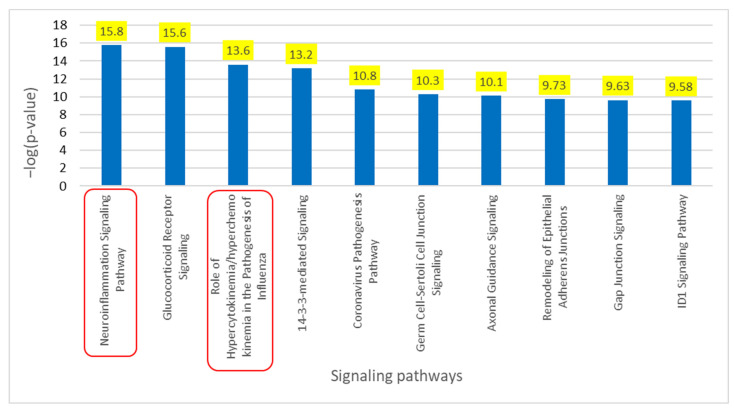
Core Analysis of the top 10 enriched canonical pathways involved in overlapping molecules between COVID-19 and *SNCA*. Core Analysis of the overlapping molecules associated with COVID-19 and *SNCA* revealed the NISP (*p*-value of 1.72 × 10^−16^) as the top enriched signaling pathway followed by the “Role of hypercytokinemia/hyperchemokinemia in the Pathogenesis of Influenza”, also known as cytokine storms (*p*-value of 2.51 × 10^−14^) as shown in the red square.

**Figure 6 ijms-24-13554-f006:**
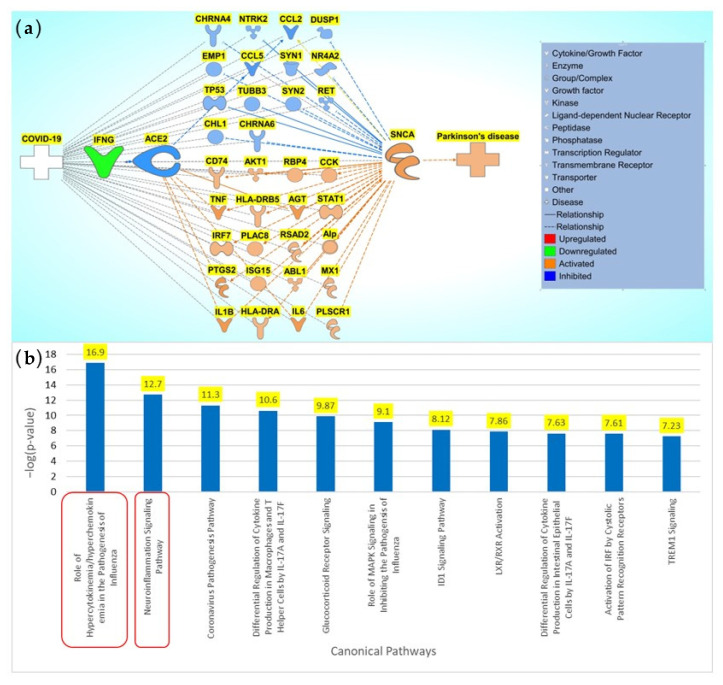
Influence of COVID-19 on *SNCA* expression and PD. (**a**) Network connectivity pathway map of the downstream effects of simulated SARS-CoV-2 infection on predicted activity of *SNCA* and PD. COVID-19-mediated downregulation of IFNG leads to the downregulation of ACE2. The downregulation of ACE2 affects the 34 intermediary molecules (14 predicted inhibited molecules (CCL2, CCL5, CHL1, CHRNA4, CHRNA6, DUSP1, EMP1, NR4A2, NTRK2, RET, SYN1, SYN2, TP53, and TUBB3) and 20 predicted activated molecules (ABL1, AGT, AKT1, Alp, CCK, CD74, HLA-DRA, HLA-DRB5, IL6, IL1B, IRF7, ISG15, MX1, PLAC8, PLSCR1, PTGS2, RBP4, RSAD2, STAT1, and TNF)) associated with COVID-19 and SNCA. The downregulation of ACE2 leads to an overall predicted activation of *SNCA* and subsequent predicted activation of PD pathogenesis. (**b**) Core Analysis of the top 10 enriched canonical pathways involved in the network connectivity pathway map of the downstream effects of simulated SARS-CoV-2 infection on predicted activity of *SNCA* and PD Core Analysis revealed the top two most-significant signaling pathways involved in the modulation of COVID-19 on *SNCA* to be the NISP (*p*-value of 1.26 × 10^−17^) and the role of hypercytokinemia/hyperchemokinemia, also known as “cytokine storms” (*p*-value of 1.99 × 10^−13^) as shown in the red squares.

**Figure 7 ijms-24-13554-f007:**
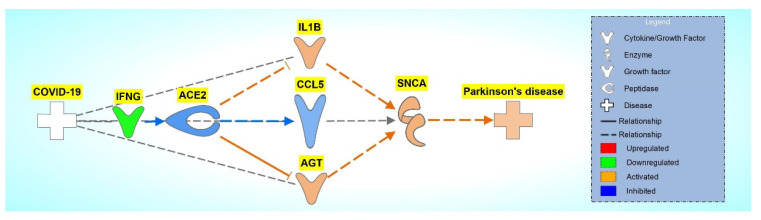
Network connectivity pathway map of ACE2’s downstream effects of simulated SARS-CoV-2 infection on predicted activity of *SNCA* and PD. COVID-19-induced inhibition via downregulation of IFNG inhibits ACE2, leading to activation of IL1B and AGT and inhibition of CCL5, causing an overall activation of *SNCA*.

**Figure 8 ijms-24-13554-f008:**
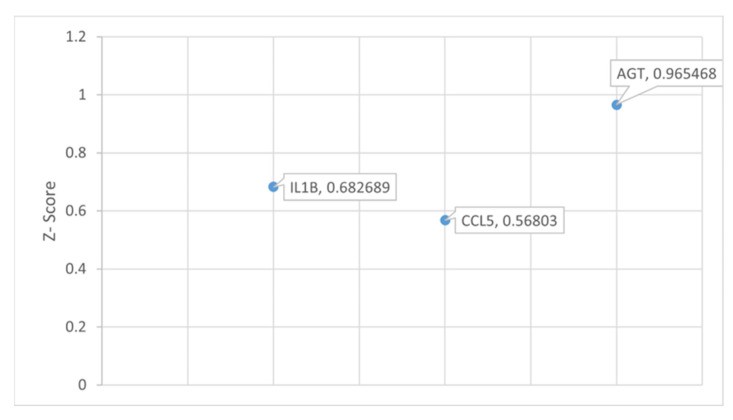
Downstream effect analysis of molecules downstream of ACE2 and upstream of *SNCA* in the network connectivity pathway map of ACE2’s downstream effects of simulated SARS-CoV-2 infection on predicted activity of *SNCA* and PD. Downstream effect analysis was performed on the intermediary molecules computing an overall z-score of 2.216187 and an overall *p*-value of 0.026685.

**Figure 9 ijms-24-13554-f009:**
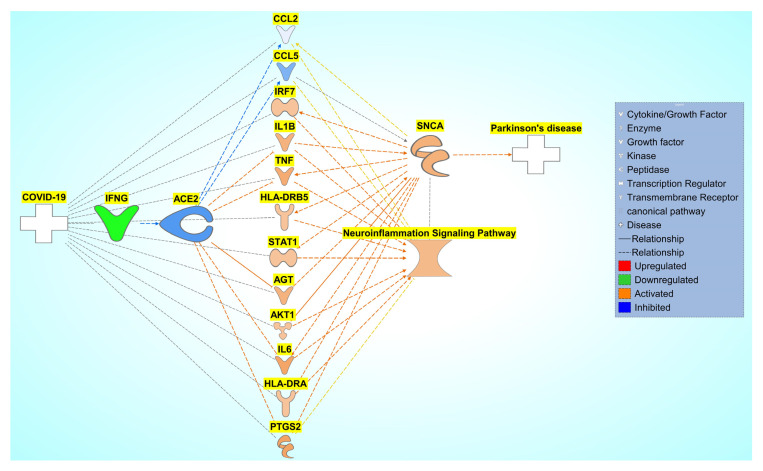
Network connectivity pathway map of the downstream effects of simulated SARS-CoV-2 infection on the predicted activity of the NISP. COVID-19-induced inhibition of ACE2 upregulates IRF7, IL1B, TNF, HLA-DRB5, STAT1, AGT, AKT1, IL6, HLA-DRA, and PTGS2 and inhibits CCL2 and CCL5, causing increased activity of the NISP and an overall increase in *SNCA* expression, eventually leading to PD progression.

**Figure 10 ijms-24-13554-f010:**
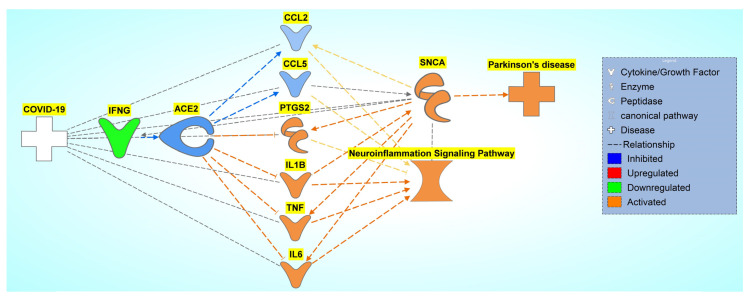
Network connectivity pathway map of ACE2’s downstream effects of simulated SARS-CoV-2 infection on predicted activity of the NISP. COVID-19-induced inhibition of ACE2 upregulates IRF7, IL1B, TNF, HLA-DRB5, STAT1, AGT, AKT1, IL6, HLA-DRA, and PTGS2 and inhibits CCL2 and CCL5, causing increased activity of the NISP and an overall increase in *SNCA* expression, eventually leading to PD progression.

**Figure 11 ijms-24-13554-f011:**
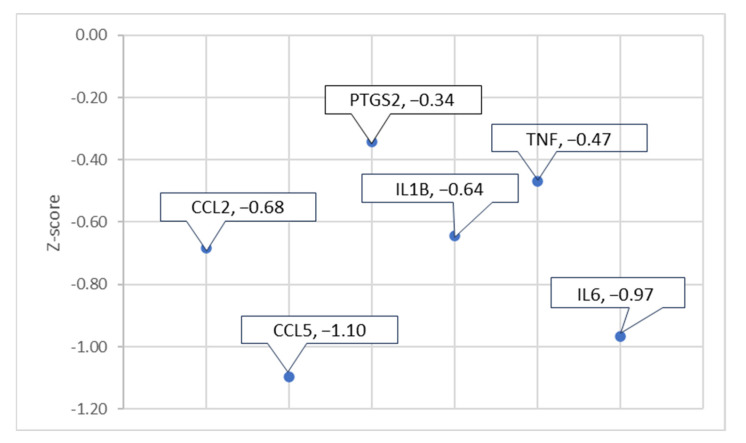
Downstream effect analysis of molecules downstream ACE2 and upstream of the NISP in the network connectivity pathway map of ACE2’s downstream effects of simulated SARS-CoV-2 infection on predicted activity of *SNCA* and PD. Downstream effect analysis was performed on the intermediary, computing an overall *p*-value of 0.000027 and an overall z-score of −4.199345911.

## Data Availability

The data presented in this study are available in this article and the given Appendix A.

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
