# Peer review of "Meta-Analysis of the Mechanisms Underlying COVID-19 Modulation of Parkinson’s Disease"

_ijms, 2023, doi:10.3390/ijms241713554_

Round 1

Reviewer 1 Report

Review of a manuscript “Meta-Analysis of the Mechanisms Underlying COVID-19 Modulation of Parkinson’s Disease”by Jonathan Zhang and coauthors

The COVID-19 pandemic has created a major global health risk due to high rate of transmission and severe complications.One of the consequences of pandemic is systemic inflammations known as a “cytokine storm”, which was often associated with neuroinflammation. In many cases the CNS of patients became damaged accompanied by blood-brain barrier disruption, vascular damage, widespread endothelial cell activation and many other downstream negative effects. The authors investigated the effect of COVID-19 on intracellular pathways analysis, targets and regulators involved in reaction of human organism on COVID-19 infections. This s a very important area of investigation and the results presented in the manuscript will be interesting for the readers of the journal. The following changes and additions should be made.

Abstract

Line 16 “… which is characterized by the formation of Lewy Bodies made of α-Synuclein protein encoded by synuclein alpha (SNCA)” Lewy bodies contain other components in addition to synuclein, so it would be more correctly to say : “… which is characterized by the formation of Lewy Bodies made primarily of α-synuclein protein encoded by synuclein alpha (SNCA) gene”.

Lines 17-18: ”This study presents the pathways involved in the downregulation of ACE2 following SARS-CoV-2 infection and its effect on PD progression”. The sentence should be corrected as follows:” This study presents the analysis of pathways involved in the downregulation of ACE2 following SARS-CoV-2 infection and its effect on PD progression”.

Introduction

Line 65 “Parkinson’s disease (PD) is a neurodegenerative disorder that leads to bradykinesia, tremors, and muscular rigidity.”

The authors should add a reference to a recent review on PD:” Emamzadeh FN et al., Parkinson’s disease: Biomarkers, Treatment, and Risk Factors. Frontiers in Neuroscience, Neurodegeneration, 12, 61230, August 2018. https://doi.org/10.3389/fnins.2018.00612

Line 69 “SNCA mutation may lead to improper translation resulting in the misfolding of the protein’s three-dimensional shape” The sentence should be corrected as follows:” SNCA mutations may lead to improper amino acid sequence resulting in the misfolding of the protein’s three-dimensional structure”.

Line 73. “SARS-CoV-2 N-protein possesses an affinity for α-Synuclein (αS)” The authors should be consistent about abbreviation od alpha-synuclein: ”SNCA or αS”?

Line 89: ”However, about 5-10 % of PD patients show early onset, below 50.” The sentence should be corrected as follows:” However, about 5-10 % of PD patients show early onset, at age below 50”.

Results.

Figure 3. SNCA is assign here as “enzyme”, but it does not possess enzymatic activity.

Discussion

Line 445. The present study was designed to understand the mechanisms in which COVID-19 affects PD through the use of IPA. The sentence is misleading. It can be corrected, for example as follows:” The present study was designed to understand using IPA the mechanisms in which COVID-19 affects PD”.

Lines 446-447: ”A set of molecules was identified between COVID-19 and PD as well as COVID-19 and SNCA where core analysis identified the top signaling pathways and upstream regulators.” The sense of this sentence is unclear. Do the authors wand to say: ”A set of interacting molecules was identified between COVID-19 and PD as well as COVID-19 and SNCA where core analysis identified the top signaling pathways and upstream regulators”?

Reviewer 2 Report

Introduction:

The introduction is really clear, and provide sufficient background. I would suggest to clear state the aims and hypothesis of this article.

Results: 

Results are well presented and not misleading. 

I would suggest to enlarge figure 6a. it is hard to read.

Materials and methods:

Do you know if the patients were vaccinated or not ? I don't see the information. THis could be interesting to mentioned it and maybe discuss it. 

Overall the article is well structured. No major flaws, I only have minor comments. 

Reviewer 3 Report

In their paper entitled “Meta-Analysis of the Mechanisms Underlying COVID-19 Modulation of Parkinson’s Disease”, the Authors report that infection by COVID-19 causes downregulation of  interferon-gamma (IFNG), thus also leading to neuroinflammation, and to Parkinson’s disease. In particular, they found that COVID-19 induces an increase in alpha synuclein expression possibly leading to its aggregation, and formation of Lewy bodies.

 Although the paper is only based on predictor tools and simulations, the results are of interest and suitable for International Journal of Molecular Sciences.

However, I suggest a few minor modifications:

1. Although the English Language is on average acceptable, I suggest a general revision to avoid too long sentences, difficult to follow;

3. Figures with different parts (a, b, etc.) should have a general title before explaining what is represented in each part of the figure;

3. In particular, figures with independent legends, reported in different parts of the manuscript should be better indicated with completely different numbers and not with, for example, 5a, 5b etc.

Please, see the general comments to the manuscript (for the Authors)

Reviewer 4 Report

I am afraid l find this article has limited scientific and medical merit.  The authors have demonstrated a complete lack of understanding about the pathophysiology of parkinson disease and limited demonstration on the course of Covid infection.  I was surprised by the tone of the article that the authors have proven the supposition that Covid makes PD worse.  There is no direct clinical data to support this in the literature.  There have been a number of papers observing reduced mobility in PD patients with Covid infection but without pathological support, the authors need to realize that their study is purely a statistical correlation between two diseases with some overlap.  

If an elderly PD patient became infected with Covid, the primary consequence will be reduced oxygen uptake, shortness of breath which will reduce the mobility of the patient for a number of reasons.  If the patient can not move so easily, there will be muscle atrophy resulting in greater movement problems.  Associating that directly with a biochemical mechanism with Covid is a stretch.  The authors do not seem to appreciate that the Covid cytokine storm occurs a number of days after initial infection, hence the use of steroid later in the disease.  If used too early, the steroids exacerbate the disesae.  The initial form of Covid pneumonia will result in increased levels of circulating cytokines that will be to some extent in the vessels of the brain but that is where this association ends.  The pathological studies of Covid in AD brains is not definitive in terms of enhanced microglial pathology.  In my opinion, some of these studies that show enhanced microgliosis are technically flawed.

The demonstrated lack of expertise in PD pathology is noticable in this article.  PD is a lot more than synuclein aggregation, and a lot more than neuroinflammation.  This article has a reference list of 25 articles of which none of them deal with PD pathology.  A detailed demonstration of how your statistical findings correspond to pathological findings is needed.  Of these 25 articles, they do not provide much support for the conclusion of this article.

The authors need to clearly present the limitations of their study and conclusions.

Extensive rewriting is needed to present the findings.  There are a number of grammatical errors.

Round 2

Reviewer 4 Report

I do not think that these modifications have improved the presentation of the hypothesis that COVID infection affects the pathology of PD.  The title and abstract of the paper are still inappropriate (click bait to use modern terminology).

This paper would need to examine another disease with known inflammatory components, such as Alzheimer's, stroke or multiple sclerosis to show a difference between those and PD.  Similarly, the authors would need to examine another viral disease compared to PD using the same methology.  If there are no differences, it solely means that COVID causes systemic inflammation 

As mentioned previously, the authors need to show a knowledge of the pathology of PD in relation to the neuropathology of COVID to support these statistical correlations.  The focus on synuclein expression is not adequate; it is not even a neuronal specific protein.  A knowledge of other lewy body diseases is needed.  The reference list shows a lack of detailed understanding of these neuropathologies.  There are many pathological scenarios about how COVID infection, leading to lung damage and reduced oxygen levels in blood can cause neuropathology, also scenarios of how COVID infections of endothelial cells in vessels can exacerbate neuropathology but these are not proven to be specific for PD.  The authors need to discuss if they want to stick to their hypothesis

Fine